# Effect of Must Hyperoxygenation on Sensory Expression and Chemical Composition of the Resulting Wines

**DOI:** 10.3390/molecules27010235

**Published:** 2021-12-30

**Authors:** Zdenek Rihak, Bozena Prusova, Michal Kumsta, Mojmir Baron

**Affiliations:** Department of Viticulture and Enology, Mendel University in Brno, Valticka 337, 691 44 Lednice, Czech Republic; xrihak2@mendelu.cz (Z.R.); michal.kumsta@mendelu.cz (M.K.); mojmir.baron@mendelu.cz (M.B.)

**Keywords:** hyperoxygenation, polyphenolic compounds, sensory analysis, white wine

## Abstract

This paper evaluates the effect of must hyperoxygenation on final wine. Lower concentrations of caftaric acid (0.29 mg·L^−1^), coutaric acid (1.37 mg·L^−1^) and Catechin (0.86 mg·L^−1^) were observed in hyperoxygenated must in contrast to control must (caftaric acid 32.78 mg·L^−1^, coutaric acid 5.01 mg·L^−1^ and Catechin 4.45 mg·L^−1^). In the final wine, hydroxybenzoic acids were found in higher concentrations in the control variant (gallic acid 2.58 mg·L^−1^, protocatechuic acid 1.02 mg·L^−1^, vanillic acid 2.05 mg·L^−1^, syringic acid 2.10 mg·L^−1^) than in the hyperoxygenated variant (2.01 mg·L^−1^, 0.86 mg·L^−1^, 0.98 mg·L^−1^ and 1.50 mg·L^−1^ respectively). Higher concentrations of total flavanols (2 mg·L^−1^ in hyperoxygenated must and 21 mg·L^−1^ in control must; 7.5 mg·L^−1^ in hyperoxygenated wine and 19.8 mg·L^−1^ in control wine) and polyphenols (97 mg·L^−1^ in hyperoxygenated must and 249 mg·L^−1^ in control must; 171 mg·L^−1^ in hyperoxygenated wine and 240 mg·L^−1^ in control wine) were found in both the must and the control wine. A total of 24 volatiles were determined using gas chromatography mass spectrometry. Statistical differences were achieved for isobutyl alcohol (26.33 mg·L^−1^ in control wine and 32.84 mg·L^−1^ in hyperoxygenated wine), or 1-propanol (7.28 mg·L^−1^ in control wine and 8.51 mg·L^−1^ in hyperoxygenated wine), while esters such as isoamyl acetate (1534.41 µg·L^−1^ in control wine and 698.67 µg·L^−1^ in hyperoxygenated wine), 1-hexyl acetate (136.32 µg·L^−1^ in control wine and 71.67 µg·L^−1^ in hyperoxygenated wine) and isobutyl acetate (73.88 µg·L^−1^ in control wine and 37.27 µg·L^−1^ in hyperoxygenated wine) had a statistically lower concentration.

## 1. Introduction

Oxygen plays an important role in the winemaking process and has both positive and negative impacts on the product’s sensory and analytical properties. Due to their polyphenolic composition, white wines are more sensitive to oxidation and thus are more susceptible to degradation and oxidative browning. Oxidative browning can be seen in the must when processing the grapes and, subsequently, in the wine. During oxidative browning of the must, the phenolic compounds are enzymatically oxidised. Oxygen consumption in the must is mediated by two oxidative enzymes: tyrosinase and laccase. The first reaction is the oxidation of caftaric and coutaric acids, which are converted to o-quinones [1]. Quinones are highly reactive compounds and can be combined with other phenolic substances to form polymerized products.

The hyperoxygenation process involves saturating the must with oxygen, which allows oxidative enzymes to convert hydroxycinnamic acids to o-quinones, which further polymerize with other phenolic compounds to form high-molecular-weight complexes. These compounds can then be removed during conventional clarification procedures [2]. By reducing the concentration of phenolic substances, lighter and more stable wines can be produced. By precipitating the flavonoid phenols responsible for bitterness and astringency, these flavors can be removed. However, hyperoxygenation can destroy the primary aromatic substances of the grapes. This method is therefore best suited for wines whose bouquets are formed during fermentation and maturation [3].

The presence of oxygen in the pre-fermentation phase changes the aromatic profile of the resulting wine. The extent of these changes depends mainly on the grape varietal, the composition of the must and the amount of oxygen introduced. The main precursors of volatile substances with aroma perception are the esters of unsaturated fatty acids [4]. Oxidation of the must can result in higher concentrations of compounds that favorably affect the aroma of the wine, such as C6 compounds, higher alcohols, fatty acids and their acetates and ethyl esters. These compounds are responsible for freshness and the formation of fruity notes [5]. Reductive processing results in higher concentrations of ester compounds, which support the fruity, varietal-specific expression of the wine’s aromatics.

The novelty of the study was to investigate how the hyperoxygenation of grape must affects the analytical and sensory parameters of final wine produced from interspecific grape vine variety. Although the effect of hyperoxygenation is relatively well known, recently white grape vine varieties are still usually processed by the reduction method. This study provides a comprehensive view of the polyphenolic composition, aromatic profile, sensory properties and sulphur dioxide content of wine produced from hyperoxygenated must of the interspecific variety Hibernal.

The content of phenolic and aromatic substances and sensory evaluation was analyzed in the final wine as well as in the control wine produced using the reductive method. The hyperoxygenated wine was used to determine if it is necessary to protect grape must from oxidation in order to preserve the wine’s aromas and if the oxidation of the must can result in an interesting product with fewer rough phenolic substances and lower levels of sulphur dioxide (SO_2_).

## 2. Results and Discussion

### 2.1. Basic Analytical Parameters

Table 1 shows the general composition of the final wines produced by the reductive method (RED) after hyperoxygenation of the must (HOX). Both variants have the same alcohol content; the statistical differences are in residual sugar and total acidity. Although there is no statistical difference between the individual acids, the difference in residual sugar may be related to the different fermentation kinetics caused by exposure of the must to oxygen. However, the result is still a dry wine, which was the intent of this experiment. Cejudo-Bastante, et al. [6] also found little difference in general composition between their reductive and oxidative variants.

What is more interesting is the difference in the level of total SO_2_. High doses of SO_2_ were added before pressing the grapes; some of it remained in the wine [7]. There are strict regulations for the use of SO_2_ in the food industry because of its toxicity and allergenic effects [8]. However, insufficient SO_2_ during the early stages of white winemaking can potentially lead to browning in the bottle as a result of the increased concentration of phenolic substances [9].

### 2.2. Volatile Aroma Compounds

The concentrations of volatile aroma compounds are presented in Table 2. There was a total of 24 chemical substances belonging to different groups, such as higher alcohols, C6 compounds or esters. The group of higher alcohols is characterized by a pleasant floral and honey aroma with a low odor threshold. Statistical differences were achieved for isobutyl alcohol, 2-phenylethanol and 1-propanol with a higher concentration in the hyperoxygenation treatment. The results differ slightly in different studies [8,10,11] due to the huge variability in the composition of must from different grape varieties and wine regions.

The formation of higher alcohols from their precursors is promoted by the presence of oxygen, even though the oxygen in the must is consumed by oxidation enzymes and thus has no effect on the fermentation process. Nevertheless, some aromatic precursors are partially oxidized, which affects the final concentration of aromatic higher alcohols [12,13].

Fermentation esters are largely responsible for wine fruitiness and play an important role in the sensory composition of young red and white wines [4]. A higher concentration of acetic esters was found in the reductive treatment, especially isoamyl acetate, 1-hexyl acetate and isobutyl acetate. The concentration of isoamyl acetate (tropical and banana aromas) was actually twice as high in the reductive method. Similar results were obtained by Cejudo-Bastante, Castro-Vázquez, Hermosín-Gutiérrez and Pérez-Coello [6] (620 µg·L^−1^ in the hyperoxidized wine and 1330 µg·L^−1^ of isoamyl acetate in the control wine) and Lukić, Horvat, Radeka, Damijanić and Staver [11] (857.41 µg·L^−1^ in heavily oxygenated grape juice and 1639.48 µg·L^−1^ of isoamyl acetate in grape mash pressing).

Other statistically significant differences were achieved in groups of ethyl esters of the medium-chain fatty acids. These esters have more interesting aromas than the others. Hexanoate has a floral, green-apple aroma. Ethyl decanoate has a soap-like odor. In white winemaking, the production of these esters can be increased by lowering the fermentation temperature and increasing must clarification [14]. For the hyperoxygenated variant, there were decreases in ethyl butyrate and ethyl hexanoate and increases in ethyl decanoate. Decreases in the concentration of this group were also noted by Cejudo-Bastante, Castro-Vázquez, Hermosín-Gutiérrez and Pérez-Coello [6] and Lukić, Horvat, Radeka, Damijanić and Staver [11]. Interestingly, the results in similar work by Cejudo-Bastante, Hermosín-Gutiérrez, Castro-Vázquez and Pérez-Coello [8], showed a higher concentration of these substances. Moio, et al. [15] achieved lower volatile acidity and higher concentrations of medium-chain fatty acid ethyl esters and acetates with high antioxidant must protection.

### 2.3. Polyphenolic Compounds

In this study, substances from the groups of flavanols (flavan-3-ols) and phenolic acids were analyzed by HPLC in must and final wines (Table 3). The concentration of total polyphenols and total flavanols in final wines was determined spectrophotometrically.

In grape must, enzymatic oxidation is largely correlated with the content of hydroxycinnamates (Figure 1), such as caftaric acid and coutaric acid, and is promoted by flavanols. Caftaric acid and coutaric acid are oxidized by polyphenol oxidases to produce *o*-quinones, which are powerful oxidants that are able to oxidize other compounds [16]. Therefore, when comparing musts, we see the largest differences in concentrations of caftaric acid, coutaric acid and catechin. The exception was GPR 2, which had a statistically higher concentration in hyperoxygenated must. This could be caused by the activity of the *laccase* enzyme in rotten grapes. Higher amounts of grape reaction product in must oxidation variants were also noted in previous studies [10,11]. Similar changes can be seen in the comparison of the final wine samples.

There were also statistical differences in tyrosol levels, with higher concentrations in reductive wine. This compound is not modified by oxygen since it is present only as a result of the fermentation of tyrosine [17]. However, oxygen saturation of the must can increase fermentation kinetics, which results in a lower tyrosol concentration in such wines. Romboli, et al. [18] discovered that slow fermentation kinetics led to higher levels of hydroxytyrosol and tyrosol.

Hydroxybenzoic acids are not as sensitive to oxidation as hydroxycinnamic acids. These are a minor component in young wines after fermentation. Gallic acid appears in the wine after standing for at least a few months as a result of the hydrolysis of the gallate esters of hydrolysable tannins and condensed tannins [19]. Figure 2 shows the differences in the selected hydroxybenzoic acids in the must and wine samples.

However, a comparison of the must and the finished wine in one variant showed interesting results. After vinification, the number of phenolic compounds increased in the hyperoxygenated variant (with the exception of vanillic acid), while hydroxycinnamic acid decreased in the reductive variant. Studies by Tian, et al. [20] and Gil-Muñoz, et al. [21] have also shown an increase in phenolic acids and flavanol compounds during fermentation.

One explanation for this could be the must clarification process. If the must is insufficiently clarified, some precipitated phenols stay in it and, subsequently, in the wine. Grape pigments are more or less soluble in an alcoholic medium but are insoluble in must [2]. Thus, the dissolved pigment can increase the concentration of hydroxycinnamic acids in hyperoxygenated wine. Moreover, SO_2_ may react with quinones, reducing it to its original phenolic compound [5].

On the other hand, the reductive variant contained much more hydroxycinnamic acid in the must protected by high doses of SO_2_. These compounds could subsequently be oxidized through a non-enzymatic (chemical) oxidation process in the wine. In chemical oxidation, oxygen radicals cause the oxidation of phenolic compounds to quinones [12].

Flavanols are found in the solid parts of the grapes and are abundant in red wine, while relatively low levels are found in white wine [20]. Figure 3 shows the same trend of increasing between must and wine samples with the lowest level of total flavanols in the hyperoxygenated must sample with connection to the enzymatic oxidation.

A comparison of total polyphenol concentration in Figure 4 shows a 30% decrease between the reductive and hyperoxygenated final wine samples. Motta, et al. [22] found that pressing the grapes in the presence of oxygen caused a significant decrease in the average concentration of total polyphenols (−23.7%) and catechins (−36.3%).

### 2.4. Sensory Analysis

Previous results of analytical measurements have already pointed out the differences between the reductive and hyperoxygenated variants. However, it was necessary to prove the impact of these two different processes on sensory expression. Hibernal wine is similar to Riesling or Sauvignon blanc, these are full-bodied wines with higher acidity and rich flavors. In Hibernal’s aroma and flavor we can look for peach, grapefruit, currant, elderflower or pineapple [23]. Figure 5 and Figure 6 show attributes selected by descriptive sensorial analysis to define the samples, together with the mean scores for each one.

Figure 5 shows more citrus and tropical fruit aroma descriptors from the reductive variant due to its higher concentration of esters, especially isoamyl acetate, which has a banana aroma. Lukić, Horvat, Radeka, Damijanić and Staver [11] and Cejudo-Bastante, Castro-Vázquez, Hermosín-Gutiérrez and Pérez-Coello [6] also noted a higher concentration of isoamyl acetate in their reductive variant. The reductive variant also had more herbal and grassy notes, which correlate to C6 unsaturated alcohol concentration. On the other hand, the hyperoxygenated variant had an aroma of candied fruit, possibly due to a higher concentration of ethyl octanoate, which smells like raisins, or 2-phenyl ethanol, with its pleasant honey scent.

Nevertheless, the hyperoxygenated variant was impressed with pure impression because the reductive variant had a stronger reductivity, as can be seen in Figure 6. The reductive variant stood out in the tasters’ notes with its distinctive tropical aromas (grapefruit, kiwi, banana); on the other hand, the hyperoxygenated variant delighted the tasters with its delicate and harmonious aroma.

Figure 6 shows that the reductive variant was fruitier, which was in line with its high concentration of esters. Herbaceous notes were also more prominent in the reductive variant, which had green, herbal, even bitter flavors, according to the tasters’ notes. In comparison, the hyperoxygenated variant was softer, which was caused by a decrease in the concentration of phenolic compounds. As Schneider [2] mentioned in his work, flavonoids have been proven to be primarily responsible for the development of bitterness and astringency. With this in mind, the tasters described the hyperoxygenated variant as smooth and rounded.

The effect of the oxygen with which the must was saturated before the start of the alcoholic fermentation is not negligible for the resulting aromatic profile. In addition to the aromatic and phenolic substances that were studied in this paper, other substances, such as thiol compounds and organic acids produced by yeast metabolism, also affect wine. For example, the nutritional stress of yeasts, which can be caused by the absence of oxygen, can lead to the production of succinic acid, thiol compounds and other volatile compounds, which can result in a final difference in the taste of wine [24].

## 3. Materials and Methods

### 3.1. Experimental Design

The experiment was performed at Mendel University in the Czech Republic using the Hibernal grape variety from Lednice’s (the Moravian wine region) 2020 harvest. The grapes were handpicked during the optimal ripening stage (pH 3.20, total acidity 9.94 g·L^−1^, °NM 20.8) following proper sanitary procedures. The destemmed and crushed grapes were pressed in a WOTTLE 1200 (Wottle, Austria) pneumatic press.

The first variant was pressed in reductive conditions with 50 mg·L^−1^ of SO_2_ as K_2_S_2_O_7_ added directly to the press machine. After pressing, the must was pumped into a 600 L steel tank. After 24 h of spontaneous sedimentation, the must was racked from the sludge. It was then inoculated with the active dry wine yeast *Saccharomyces cerevisiae* (Vitiferm Alba Fria BIO, 2B FermControl Germany). After fermentation, the wine was racked and supplied with 40 mg·L^−1^ of SO_2_ as K_2_S_2_O_7_. The free SO_2_ content was maintained at a level of 25–30 mg·L^−1^ during winemaking. A dose of 100 g per 1 hL of sodium-calcium bentonite was used for clarification. After sedimentation, a Filtro (Enotech srl, Italy) flat sheet filter size 20 × 20 cm was used for filtration.

The second (hyperoxidized) variant was pressed without using SO_2_. After pressing, the must was racked into a 600 L steel tank and hyperoxygenated. A micro-oxygenation device, the Oxy Genius Plus (Parsec srl, Italy), was used for must oxidation. A silicon diffuser was connected to an oxygen bottle and placed in the bottom of the tank. The oxygen dose was set at 50 mg·L^−1^ over five hours. The rest of the procedures were the same as for the reductive variant. Samples were collected and analyzed from both batches of the must and the final wines.

### 3.2. Basic Chemical Parameters

The sugar concentration of the grape must was analyzed with an ATAGO PAL-1 (Atago, Tokyo, Japan) refractometer. The Brix scale was converted to °NM.

The pH value of the must was estimated using a WTW 526 pH meter (WTW, Weilheim, Germany) with a SenTix 21 pH electrode.

The total acidity of and assimilable nitrogen in the must were estimated using a TITROLINE EASY (SI Analytics GmbH, Mainz, Germany) automatic titrator. A 0.1 mol·L^−1^ solution of NaOH was used as a titration reagent. For the analyses, 10 mL wine samples diluted with 10 mL of distilled water were used. Individual samples were thereafter titrated up to pH 8.1, again using the SenTix 21 pH electrode. After titration, the consumption of the NaOH solution was read on the titrator’s display. This consumption was multiplied by the factor of the NaOH solution used for the titration with a coefficient of 0.75. The result was equal to the content of titratable acids in the wine sample (mg·L^−1^). After titration, 5 mL of formaldehyde was added; the pH value declined, so the sample was again titrated to a pH of 8.1. The assimilable nitrogen was calculated from the second NaOH consumption value; the result was expressed in mg·L^−1^ [25].

The basic parameters of the resulting wine (alcohol, pH, residual sugar, titratable acids, malic, lactic, tartaric, acetic, acids and glycerol) were determined with an Alpha FTIR analyzer (Bruker, Bremen, Germany) using the attenuated total reflection (ATR) sampling technique. Before the first measurement, the spectrometer was thoroughly rinsed with deionized water and the background was determined using a blank sample (deionized water). For the analyses, 1 mL samples were taken with a syringe; 0.5 mL was used to rinse the system while the remaining volume of 0.5 mL was analyzed three times. The measured values were evaluated automatically using the OpusWine software (Bruker, Bremen, Germany).

### 3.3. Determination of Free and Total SO_2_

The determination was performed through iodometric titration using a standard iodine solution [25].

Determination of free SO_2_: 50 mL of wine was pipetted into a 250 mL titration flask; 10 mL of 16% H_2_SO_4_ and 5 mL of 0.5% starch were added. The wine sample was immediately titrated with 0.02 M iodine solution until the equivalence point was reached.

Calculation of free SO_2_ content, Equation (1):x = a × f × 12.8
(1)a = consumption of 0.02 M I_2_ solution;x = free SO_2_ (mg·L^−1^);f = factor of 0.02 M I_2_ solution

Determination of total SO_2_:

A 50 mL wine sample was pipetted into a 250 mL titration flask and 25 mL 1 M NaOH was added. After 15 min, 15 mL of 16% H_2_SO_4_ and 5 mL of 0.5% starch were added and immediately titrated with 0.02 M iodine solution until the equivalence point was reached.

Calculation of total SO_2_ content, Equation (2):x = a × f × 12.8 = total SO_2_ (mg·L^−1^).(2)

### 3.4. Analysis of Volatile Aroma Compounds

The concentration of the individual volatile compounds in the wine was determined according to the method of extraction using methyl tert-butyl ether (MTBE): 20 mL of wine was pipetted into a 25 mL volumetric flask along with 50 μL of 2-nonanol solution in ethanol. This compound was used as an internal standard (in a concentration of 400 mg·L^−1^) and 5 mL of a saturated (NH_4_)_2_SO_4_ solution. The flask’s contents were thoroughly stirred; 0.75 mL of the extraction solvent (MTBE with an addition of 1% cyclohexane) was then added. After another thorough stirring and the separation of individual phases, the upper organic layer was placed into a micro test tube along with the produced emulsion and then centrifuged. The clear organic phase was dried over anhydrous magnesium sulphate prior to the GCMS analysis. The extraction and subsequent GC analysis were performed three times. The average values and standard deviations were determined using Excel and Statistica 10. The determination was performed in a Shimadzu gas chromatograph GC-17A (Shimadzu, Duisburg, Germany) equipped with an autosampler (AOC-5000, Shimadzu, Duisburg, Germany) and connected to a QP mass spectrometer detector (QP-5050A, Shimadzu, Duisburg, Germany) with EI ionization.

Identification was performed using GCsolution software (LabSolutions, version 1.20, Mundelein, IL, US). Analysis was performed under the following conditions of separation: column: DB-WAX 30 m × 0.25 mm; 0.25 μm stationary phase polyethylene glycol). The voltage of the detector was 1.5 kV. The sample injection volume was 1 μL, with a split ratio of 1:5. The flow of the carrier gas (He) was 1 mL·min (linear gas velocity 36 cm/s) and the temperature of the injection port was 180 °C. The initial column temperature was 45 °C maintained for 3.5 min, followed by temperature gradients: to 75 °C gradient 6 °C·min, to 126 °C gradient 3 °C·min^−1^, to 190 °C gradient 4 °C·min^−1^ and to 250 °C gradient 5 °C·min^−1^. The final temperature was subsequently maintained for 6.5 min. The total length of analyses was 60 min. The detector worked in SCAN mode in 0.25-s intervals with a range of 14–264. Individual compounds were identified by comparing the MS spectrum and the retention time with the NIST 107 library. The identity of the substances and the validity of the method were verified by a standard addition of test substances. This procedure was also used to quantify the analytes [26].

### 3.5. Determination of Total Polyphenols Concentration

The Folin–Ciocalteu method was used to determine total polyphenolic compounds. All samples were analyzed in triplicate; the resulting value was obtained as the average of these measurements.

A 40 µL sample was pipetted into a cuvette (3 mL) and diluted with 1960 µL of distilled water. Subsequently, 50 µL of Folin–Ciocalteu reagent was added to the cuvette. The mixture was then shaken thoroughly. After three minutes, 300 µL of 20% Na_2_CO_3_ decahydrate solution were added. The reaction mixture was shaken and incubated at 22 °C for 120 min. Absorbance was measured using a double-beam spectrophotometer (SPECORD 210, Carl-Zeiss, Jena, Germany) at λ = 750 nm against a blank sample. The results were expressed as a gallic acid equivalent [27,28]

### 3.6. Determination of Total Polyphenols Flavanols

The concentration of total flavanols was determined by a method based on their reaction with *p*-dimethylaminocinnamaldehyde (DMACA). This method, in contrast to the widely used reaction with vanillin, does not interfere with anthocyanins. In addition, it provides higher sensitivity and selectivity. A sample of 10 μL was added to 240 μL of reagent (0.1% DMACA and 300 mM HCl in MeOH). The reaction time was 600 s, after which the absorbance at 620 nm was measured. The concentration of total flavanols was determined on the basis of a calibration curve using epicatechin as a standard (10–200 mg·L^−1^). The results are expressed as mg·L^−1^ equivalents of epicatechin [29].

### 3.7. Determination of Individual Phenolic Compounds by HPLC

The selected polyphenolic compounds were determined by unpublished method with direct injection of sample. Wines were centrifuged (3000× *g*; 6 min) and diluted by 100 mM HClO_4_. Wines were diluted with ratio 1:1 for white.

The chromatographic system Shimadzu LC-10A (Shimadzu, Duisburg, Germany) consisted of two pumps LC-10ADvp, column thermostat with manual injection valve, DAD detector SPD-M10Avp and a personal computer running the chromatographic software LCsolution. The chromatographic separations were performed on a column Alltech Alltima C18 (3 μm, 3150 mm, Grace, Deerfield, IL, USA) equipped with a guard column (3 × 7.5 mm^2^) filled with the same sorbent. The temperature of separations was 60 °C. The mobile phases were the following: A = 15 mM HClO_4_ and B = 15 mM HClO_4_, 10% MeOH, 50% ACN.

The gradient program is described below and the flow rate was 0.6 mL·min^−1^:
0.00 min4% B20.00 min28% B30.00 min42% B35.00 min60% B38.00 min100% B40.00 min100% B40.01 min0% B40.99 min0% B41.00 min4% B43.00 min4% B

The total length of the analysis is 43 min. Regeneration time is 4 min. Data recorded in the range 200–520 nm.

Determination of individual components on the basis of calibration curves of standards:200 nm: Catechin; epicatechin260 nm: vanillic acid; protocatechuic acid; 4-hydroxybenzoic acid275 nm: gallic and syringic acid285 nm: Cis-piceid; cis-resveratrol310 nm: *p*-coumaric acid and its derivatives; trans-piceid; trans-resveratrol325 nm: caffeic acid and its derivatives; ferulic acid and its derivatives; piceatannol360 nm: rutin; myricetin; quercetin; kaemferol; isorhamnetin520 nm: anthocyanins

Derivatives of hydroxycinnamic acids were calibrated on basic acids from which they are derived. Anthocyanins have been calibrated to malvidin-3,5-diglucoside.

### 3.8. Reagents and Standard Solutions

Acetonitrile (ACN) and methanol (MeOH) were HPLC supergradient purity. Catechin, epicatechin, vanillic acid, protocatechuic acid, 4-hydroxybenzoic acid, gallic acid, syringic acid, *p*-coumaric acid, *trans*-resveratrol, *trans*-piceid, caffeic acid, ferulic acid, piceatannol, rutin, myricetin, quercetin, kaemferol, isorhamnetin, and perchloric acid were obtained from Sigma Chemical Co. (St. Louis, MO, USA). Malvidine -3,5-diglukoside was purchased from Indofine Chemical Company. Inc. (Hillsborough, NJ, USA).

Other used chemicals were at least analytical grade and were obtained from local suppliers (Lachema, Penta, Prague, Czech Republic).

A stock standard solution was prepared by accurately weighting about 10 mg of each phenol in 25 mL volumetric flask. The standard was dissolved in 10 mL of acetonitrile, bringing up to volume with distilled water [27].

### 3.9. Sensory Analysis

Sensory evaluation was carried out by a panel of twelve professional tasters in the sensory analysis laboratory of the Mendel University in Brno. The tasting room was designed to conduct sensory analyses under known and controlled conditions as described in the ISO 8589 standards. The wines were blind tasted in clear Institut national de l’origine et de la qualité (INAO) glasses by eight qualified assessors, also in accordance with ISO 8586 standards. A descriptive analysis was carried out to evaluate eleven types of descriptive attributes of the wines. Evaluation of the aromatic and mightiness profiles concerned three pairs of opposite characteristics: oxidative versus reductive, herbal versus soft and waxy versus fruity. For each pair, the more pronounced characteristic was chosen and rated on a scale of 1–5. The graph shows the resulting structure of the wines. The average of all the final ratings for each wine was calculated and graphs were created to reflect the results. The evaluation was quantified using a scale with an unstructured intensity of 10 points.

Each taster evaluated two samples per session. The test room had individual, white, illuminated booths, and samples were served individually and coded in tasting glasses (ISO) each containing 50 mL of wine at 18 ± 2 °C.

### 3.10. Statistical Analysis

Statistical analysis and graphs were created using MS Excel 2010 (Microsoft Office, Redmond, WA, USA) and Statistica 10 (Copyright © StatSoft). A one-way analysis of variance (ANOVA) and Fischer’s least significant difference (LSD) test were used to compare the means (*n* = 3) at the level of significance of *p* < 0.05.

## 4. Conclusions

The results of this study show the differences between wines from reductive grape processing and targeted must oxidation (hyperoxygenation). The Hibernal grape variety was chosen for its typical aroma profile. The experiment’s main goal was to make wine without using SO_2_ during grape processing, since sulphur is an allergen and drinking wine with a high concentration of total SO_2_ can cause headaches. Moreover, there could be problems due to the maximum doses allowed by regulatory entities.

Another useful goal was to affect the sensory expression of the final product. Oxygen in grape must can modify the composition of volatile aroma compounds like esters or higher alcohols. In general, there were fewer ester compounds in oxidation must variants, so this may not be the correct way to make expressive wine from a primarily aromatic grape variety. On the other hand, the concentration of phenolic compounds decreased due to the enzymatic oxidation of the must, giving the resultant wine a less astringent, softer taste. However, there are some fining agents used to remove phenolic substrates. In this case, undesirable phenols under reductive conditions can be removed without the oxidation of desirable aroma compounds. Of course, this remains at the discretion of the winemaker.

Targeted oxidation of white grape must or hyperoxygenation is a pre-fermentation processing technique. This study did not find any negative impact of hyperoxygenation on Hibernal white wine and, in addition, this study provides a comprehensive view of the polyphenolic composition, aromatic profile and sulphur dioxide content of wine made from hyperoxygenated must of the interspecific Hibernal variety, which is also the novelty of this study.

## Figures and Tables

**Figure 1 molecules-27-00235-f001:**
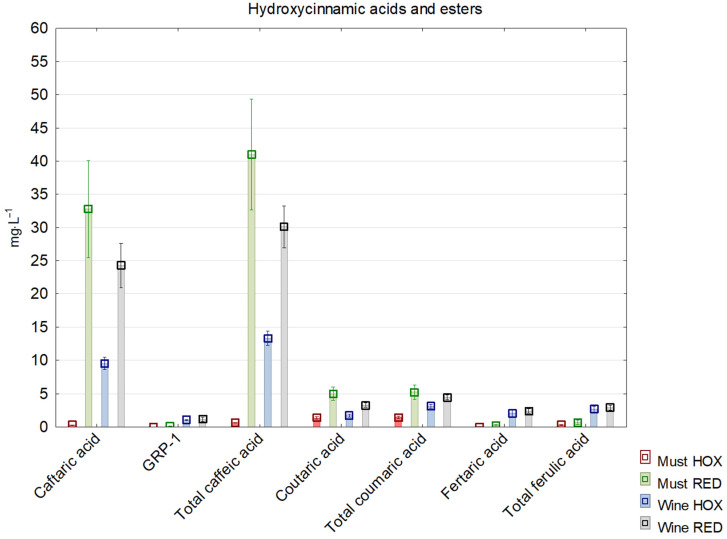
Concentration of selected hydroxycinnamic acids and esters in reductive and hyperoxidized musts and wines.

**Figure 2 molecules-27-00235-f002:**
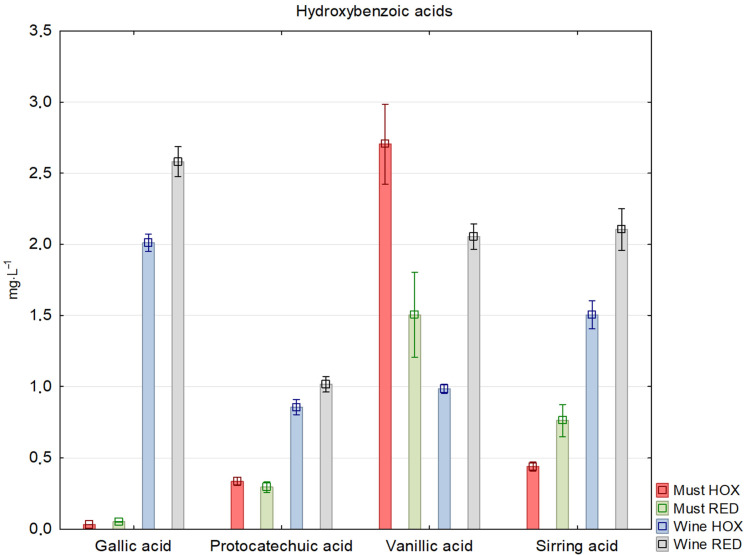
Concentration of selected hydroxybenzoic acids and esters in reductive (RED) and hyperoxidized (HOX) musts and wines.

**Figure 3 molecules-27-00235-f003:**
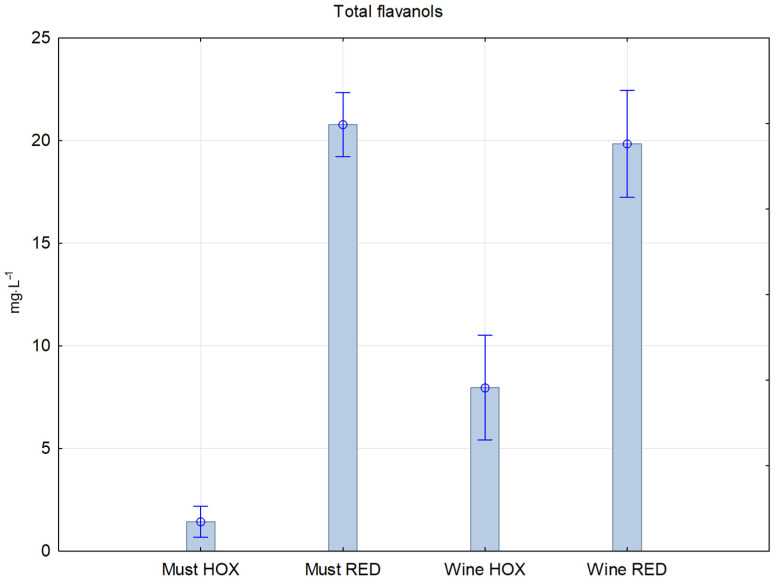
Concentration of total flavanols in reductive and hyperoxidized musts and wines.

**Figure 4 molecules-27-00235-f004:**
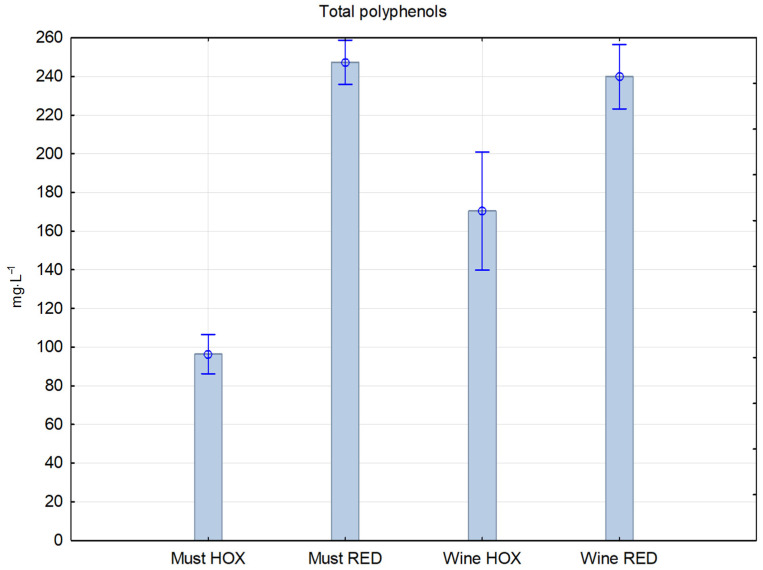
Concentration of total polyphenols in reductive and hyperoxidized musts and wines.

**Figure 5 molecules-27-00235-f005:**
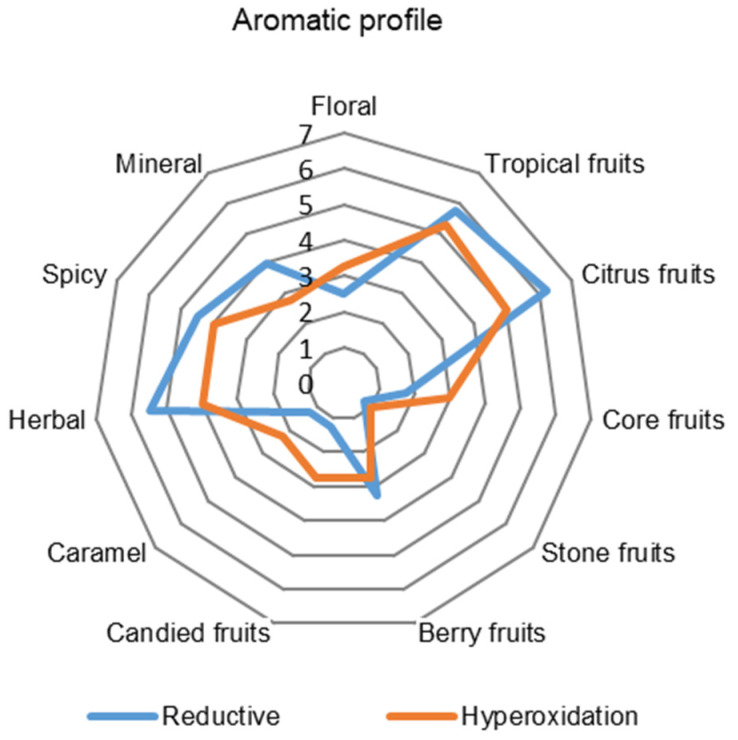
Aromatic profile comparing the results of reductive (RED) and hyperoxidized (HOX) Hibernal grape processing. The values on the graph are averages from eight evaluators.

**Figure 6 molecules-27-00235-f006:**
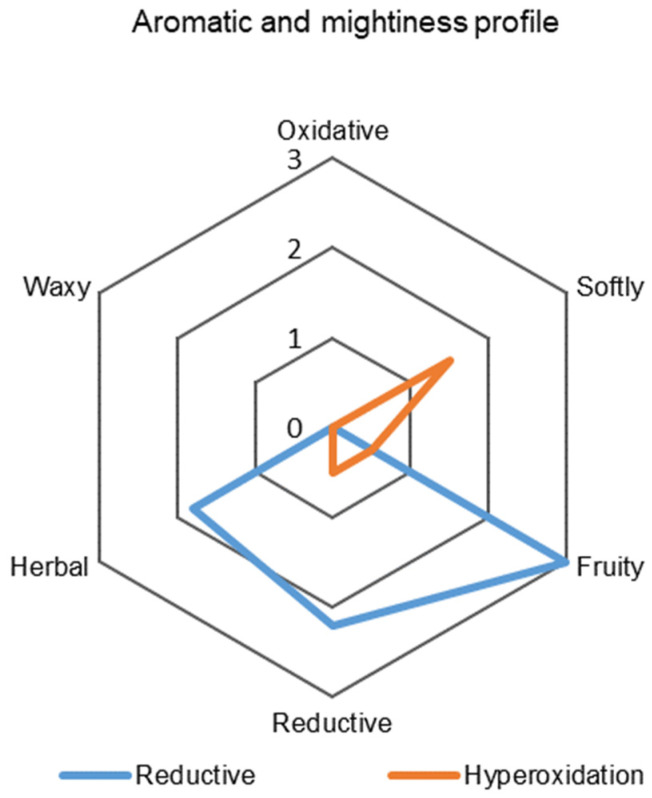
Aromatic and mightiness profile comparison of the result of reductive (RED) and hyperoxidized (HOX) Hibernal grape processing. The values on the graph are averages from eight evaluators.

**Table 1 molecules-27-00235-t001:** Basic analytical parameters of final wines.

Wine	RED	HOX
Alcohol % vol	12.57 ± 0.08	12.79 ± 0.07
Total acidity g·L^−1^	8.12 ± 0.11 a	9.45 ± 0.19 b
Residual sugar g·L^−1^	1.37 ± 0.37 a	0.42 ± 0.59 b
pH	3.17 ± 0.02 a	3.02 ± 0.02 b
Malic acid g·L^−1^	3.20 ± 0.23	3.62 ± 0.40
Lactic acid g·L^−1^	0.36 ± 0.09	0.27 ± 0.15
Acetic acid g·L^−1^	0.41 ± 0.05	0.42 ± 0.01
Tartaric acid g·L^−1^	3.35 ± 0.20	3.84 ± 0.30
Glycerol g·L^−1^	7.02 ± 0.86	7.14 ± 0.10
Free SO_2_ mg·L^−1^	24.6 ± 0.33 a	26.7 ± 0.33 b
Total SO_2_ mg·L^−1^	102.7 ± 0.88 a	69.0 ± 0.58 b

Note: The average values (*n* = 3) were combined by contribution to homogeneous groups according to Fisher’s Least significant difference (LSD) test, where different letters in the same row indicate significant differences between RED and HOX (α = 0.05).

**Table 2 molecules-27-00235-t002:** Concentrations of volatile aroma compounds in final wines.

Volatile Compounds		Aroma Descriptor *	RED	HOX
**Higher alcohols**				
Isoamyl alcohol	mg·L^−1^	ripe fruit	270.25 ± 4.88	277.74 ± 5.92
Isobutyl alcohol	mg·L^−1^	ether, fruits	26.33 ± 0.24 a	32.84 ± 0.56 b
2-Phenylethanol	mg·L^−1^	rose, talc, honey	14.73 ± 0.12 a	25.82 ± 0.35 b
1-Propanol	mg·L^−1^	fruits, alcohol	7.28 ± 0.04 a	8.51 ± 0.20 b
1-Hexanol	mg·L^−1^	fresh cut grass	1.23 ± 0.03	1.14 ± 0.02
**C6 unsaturated alcohols**				
(E)-3-Hexen-1-ol	µg·L^−1^	grass	43.07 ± 1.88 a	19.50 ± 1.03 b
(Z)-3-Hexen-1-ol	µg·L^−1^	grass	32.06 ± 1.90 a	14.44 ± 0.95 b
**Acetic esteres**				
Ethyl acetate	mg·L^−1^	fruity, nail polish	23.02 ± 0.48 a	33.28 ± 0.51 b
Isoamyl acetate	µg·L^−1^	banana	1534.41 ± 31.87 a	698.67 ± 28.64 b
Hexyl acetate	µg·L^−1^	pear	136.32 ± 1.82 a	71.67 ± 1.08 b
2-Phenylethyl acetate	µg·L^−1^	peaches, honey, roses	170.10 ± 0.72	168.95 ± 1.91
Isobutyl acetate	µg·L^−1^	fruits	73.88 ± 0.49 a	37.27 ± 0.51 b
**Ethyl esteres**				
Ethyl butyrate	µg·L^−1^	Fruits	191.15 ± 5.63 a	150.80 ± 2.29 b
Ethyl hexanoate	µg·L^−1^	Flowers, green apple	459.38 ± 9.45 a	347.04 ± 1.44 b
Ethyl oktanoate	µg·L^−1^	Raisins	781.82 ± 10.97	806.54 ± 15.80
Ethyl decanoate	µg·L^−1^	flowers, soap-like	221.86 ± 4.45 a	295.39 ± 1.90 b
Ethyl lactate	mg·L^−1^		6.55 ± 0.21 a	12.22 ± 0.22 b
Diethyl succinate	mg·L^−1^	melon, vinous	0.25 ± 0.02 a	0.69 ± 0.01 b
Diethylmalate	mg·L^−1^		1.12 ± 0.08 a	2.52 ± 0.09 b
**Volatile phenols**				
4-Vinylguaiacol	µg·L^−1^	smoky, spicy	50.62 ± 0.67 a	26.68 ± 1.85 b
4-Vinylfenol	µg·L^−1^	almond shell	216.50 ± 5.87 a	137.62 ± 5.95 b
**Others**				
Acetoin	mg·L^−1^	buttery, cream	0.55 ± 0.07	0.47 ± 0.04
2,3-Butandiol	mg·L^−1^		335.30 ± 8.79 a	677.92 ± 28.22 b
Benzaldehyde	µg·L^−1^	bitter, cherry	9.40 ± 0.80 a	17.60 ± 0.75 b

Note: The average values (*n* = 3) were combined by contribution to homogeneous groups according to Fisher’s Least significant difference (LSD) test, where different letters in the same row indicate significant differences between RED and HOX (α = 0.05). * Aroma descriptors reported in the literature [11].

**Table 3 molecules-27-00235-t003:** Mean value of concentration of selected phenolic compounds in musts and final wines (mg·L^−1^).

Phenols (mg·L^−1^)	HOX		RED	
	Must ^1^	Final Wine	Must ^2^	Final Wine
**Hydroxybenzoic acids**				
Gallic acid	0.03 a	2.01 ± 0.01 b	0.05 a	2.58 ± 0.02 c
Protocatechuic acid	0.34 ± 0.01 a	0.86 ± 0.01 b	0.30 ± 0.01 a	1.02 ± 0.01 c
Vanillic acid	2.70 ±0.07 a	0.98 ± 0.01 b	1.50 ± 0.07 c	2.05 ± 0.02 d
Sirring acid	0.44 ± 0.01 a	1.50 ± 0.02 b	0.76 ± 0.03 c	2.10 ± 0.03 d
**Hydroxycinnamic acids**				
Caftaric acid	0.29 ± 0.01 a	9.54 ± 0.21 b	32.78 ± 1.70 c	24.29 ± 0.77 d
GRP-1	0.01 a	1.04 ± 0.02 b	0.09 ± 0.01 a	1.15 ± 0.07 d
GRP-2	0.14 a	2.50 ± 0.02 b	7.89 ± 0.23 c	4.55 ± 0.05 d
Total caffeic acid	0.66 ± 0.01 a	13.29 ± 0.25 b	41 ± 1.95 c	30.10 ± 0.73 d
Coutaric acid	1.37 ± 0.05 a	1.73 ± 0.05 b	5.01 ± 0.23 c	3.29 ± 0.07 d
Total coumaric acid	1.45 ± 0.05 a	3.13 ± 0.07 b	5.25 ± 0.05 c	4.44 ± 0.08 d
Fertaric acid	0.04 a	2.09 ± 0.12 b	0.27 ± 0.02 c	2.41 ± 0.08 d
Total ferulic acid	0.31 ± 0.01 a	2.67 ± 0.11 b	0.70 ± 0.07 c	2.91 ± 0.09 b
**Flavanols**				
Catechin	0.86 ± 0.02 a	1.71 ± 0.06 b	4.45 ± 0.26 c	4.98 ± 0.13 d
Epicatechin	0.30 ± 0.01 a	1.25 ± 0.03 b	0.25 ± 0.01 a	2.71 ± 0.03 c
**Others**				
Tyrosol	0.58 a	13.44 ± 0.08 b	0.97 ± 0.01 c	18.91 ± 0.9 d

Note: The average values (*n* = 3) were combined by contribution into homogeneous groups according to Fisher’s Least significant difference (LSD) test, where different letters in the same row indicate significant differences between RED and HOX (α = 0.05). ^1^ value analysis after hyperoxygenation of must. ^2^ value analysis of must after press.

## Data Availability

Not applicable.

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
