# Peer review of "Effect of Must Hyperoxygenation on Sensory Expression and Chemical Composition of the Resulting Wines"

_molecules, 2021, doi:10.3390/molecules27010235_

Round 1

Reviewer 1 Report

In this paper, the author demonstrated the effect of must hyperoxidation on wine chemical compositions and sensory expression. Compared to the reductive process method, this manuscript discussed that hyperoxidation could affect the concentration of phenolic and aromatic compounds and sensory evaluation of the final wine. Various analytical methods have been applied to carefully determine the chemical compositions in the experiments. And statistical differences were able to achieve to tell among comparison experiments. One major goal to apply hyperoxidation is to avoid using SO2 during grape processing, which is an allergen and also cause headaches. The manuscript supported that hyperoxidation could be a potential approach to alleviate this problem in winemaking.  

Questions:

  • At the top of page 4, it would be helpful to let the readers know that the QP detector you mentioned is a mass spectrometer, and also describe the ionization mode of this source (e.g., EI) in this paragraph.
  • In the Table 2, I observed comma marks were used in the values presented in this table. I'm wondering whether those should be decimal marks or not.
  • In the page 8, line 294, there discussed that "oxygen saturation of the must can increase fermentation kinetics", which may indicate that the fermentation kinetic was faster in the HOX than in the RED. However, in the Table 1, the HOX showed a much higher residual sugar level than RED process. And it was mentioned that "residual sugar may be related to the different fermentation kinetics caused by exposure of the must to oxygen" (page 5, line 216), which seemed to hypothesize that the fermentation kinetics of HOX should be slower and cause more sugar leftover than RED due to more exposure to oxygen. The discussions of fermentation kinetics in these two places did not agree with each other. It would be much helpful the author could provide more explanation and discussion on this part.  

Author Response

Dear Reviewer, thank you for reviewing the article and valuable comments. In the text below you will find the answers to your comments.

Point 1: At the top of page 4, it would be helpful to let the readers know that the QP detector you mentioned is a mass spectrometer, and also describe the ionization mode of this source (e.g., EI) in this paragraph.

Response 1: Thank you, information was added: The determination was performed in a Shimadzu gas chromatograph (GC-17A) equipped with an autosampler (AOC-5000) and connected to a QP mass spectrometer detector (QP-5050A) with EI ionization.

Point 2: In the Table 2, I observed comma marks were used in the values presented in this table. I'm wondering whether those should be decimal marks or not.

Response 2: Thank you for noticing this error. Commas are now corrected to dots.

Point 3: In the page 8, line 294, there discussed that "oxygen saturation of the must can increase fermentation kinetics", which may indicate that the fermentation kinetic was faster in the HOX than in the RED. However, in the Table 1, the HOX showed a much higher residual sugar level than RED process. And it was mentioned that "residual sugar may be related to the different fermentation kinetics caused by exposure of the must to oxygen" (page 5, line 216), which seemed to hypothesize that the fermentation kinetics of HOX should be slower and cause more sugar leftover than RED due to more exposure to oxygen. The discussions of fermentation kinetics in these two places did not agree with each other. It would be much helpful the author could provide more explanation and discussion on this part.  

Response 3: Thank you for pointing out this relatively serious mistake in Table 1 that occurred during manuscript conceptualization. In fact, the fermentation kinetic was better in the hyperoxidized variant, as discussed. Values in Table 1 were checked and corrected. There was also mistake in concentration of sulphur dioxide.

Reviewer 2 Report

I have not found major drawbacks in the proposed manuscript. I consider it as suitable for publication. Perhaps slight language editing is advisable.

Author Response

Dear Reviewer, 

Thank you for reviewing and accepting the article, the English language has already been checked by a native speaker.

Reviewer 3 Report

The objective of the article seems interesting, but if the authors mention that the method of Hyperoxidation of the must of white wines is already well used, what is the innovation of this research?

Unfortunately the article presents many methodological problems and should not be recommended for publication in this journal, such as: (1) there are no replicates of winemaking; (2) the scientific validity of the method used to perform the classical analyzes (Alpha FTIR 103 analyser) is questionable. For the preparation of the calibration models were used wines of the same variety and region?; (3) All the volatile compounds were  only tentatively identified, so it is not possible to be sure if they are really present in the samples. Furthermore, it is clear that many of the volatile compounds found are not mentioned. Since by this type of extraction many compounds are found in the chromatographic run. Probably because splitless mode has not been evaluated. Additionally, how did the authors quantify the volatile compounds shown in Table 2? In this table, do not appear the references for the aroma notes mentioned for each compound; (4) Why dilute the wine and must 10x with mobile phase before injecting the sample on the HPLC? What is the identity of the standards used to quantify the phenolic compounds? What is the wavelength used in the detector UV-VIS? Where are the validation parameters of the method tested for quantification of phenolic compounds by HPLC? (5) The way of the sensory analysis was conducted is not valid for scientific publications. The sensorial analysis must be remake with a descriptive sensory analysis technique recognized by the scientific community (such as QDA).

Author Response

Dear Reviewer, thank you for reviewing the article and valuable comments. In the text below you will find the answers to your comments.

Point 1: The objective of the article seems interesting, but if the authors mention that the method of Hyperoxidation of the must of white wines is already well used, what is the innovation of this research?

Response 2: The novelty of the study was to investigate how the hyperoxygenation of grape must affects the analytical and sensory parameters of final wine produced from interspecific grape vine variety. Although the effect of hyperoxygenation is relatively well known, recently white grape vine varieties are still usually processed by the reduction method. This study provides a comprehensive view of the polyphenolic composition, aromatic profile, sensory properties and sulfur dioxide content of wine produced from hyperoxygenated must of the interspecific variety Hibernal.

Point 2: Unfortunately the article presents many methodological problems and should not be recommended for publication in this journal, such as there are no replicates of winemaking;

Response 2: Unfortunately, the experiment can not be repeated, but similar pilot experiments before this confirmed the results of this study.  

Point 3: the scientific validity of the method used to perform the classical analyzes (Alpha FTIR 103 analyser) is questionable. For the preparation of the calibration models were used wines of the same variety and region?;

Response 3: For the preparation of the calibration models were used wines of the same variety and the same year (grapes were obtained from vineyards of our university), calibration was prepared according to results of HPLC and reference method such a total acidity by titration method, alcohol by destilation method and individual acids and sugars by HPLC method.

Point 4: All the volatile compounds were  only tentatively identified, so it is not possible to be sure if they are really present in the samples. Furthermore, it is clear that many of the volatile compounds found are not mentioned. Since by this type of extraction many compounds are found in the chromatographic run. Probably because splitless mode has not been evaluated. Additionally, how did the authors quantify the volatile compounds shown in Table 2? In this table, do not appear the references for the aroma notes mentioned for each compound;

Response 4: The identity of the substances and the validity of the method were verified by a standard addition of test substances. This procedure was also used to quantify the analytes. The same procedure was used for the previously published method, where the recovery factor is also described: Prusova, B.; Baron, M. Effect of controlled micro-oxygenation on white wine. Ciência e Técnica Vitivinícola 2018, 33, 78-89.

Point 5: Why dilute the wine and must 10x with mobile phase before injecting the sample on the HPLC? What is the identity of the standards used to quantify the phenolic compounds? What is the wavelength used in the detector UV-VIS? Where are the validation parameters of the method tested for quantification of phenolic compounds by HPLC?

Response 5: Thank you for your comment. During the conceptualization of the article, an error occurred while writing the method for HPLC determination. A DAD detector was used for the measurement, which measured phenolic substances at different wavelengths. This is diluted so that the detector is not flooded and there are linear calibration curves - the dilution is done twice for white wines. In addition, dilution suppresses the dissociation of phenolic acids and significantly improves the separation of substances. The method is already corrected in the article.

Point 6: The way of the sensory analysis was conducted is not valid for scientific publications. The sensorial analysis must be remake with a descriptive sensory analysis technique recognized by the scientific community (such as QDA).

Response 6: Thank you, it´s already corrected.

Reviewer 4 Report

Dear Author,

  1. Please revise the abstract, so that the most important results are presented clearly, with less focus on the conclusions. Elusive expressions such as "decreased concentration" are usually not allowed. It must be specified how much the value of a parameter decreases or increases.
  2. Abbreviations that have not been previously defined cannot be used in the abstract.
  3. The OIV Code of Oenological Practices includes the method by which the must is oxygenated to improve the stability of wine (oeno 545A / 2016), so the term hyperoxidation is not used correctly. Is it hyperoxygenation?
  4. It should be specified what is the degree of novelty and relevance of the research.
  5. Line 57: “aromatic substances” refers to aromatic ring compounds or flavor compounds? Please clarify because it is ambiguous.
  6. The units of measurement must be redefined according to the OIV and corrected throughout the text.
  7. Line 62: “SO2” Chemical formulas must be corrected, indices are not placed correctly.

We recommend MAJOR Revisions (with addition of extra data to justify novelty).

Author Response

Dear Reviewer, thank you for reviewing the article and valuable comments. In the text below you will find the answers to your comments.

Point 1: Please revise the abstract, so that the most important results are presented clearly, with less focus on the conclusions. Elusive expressions such as "decreased concentration" are usually not allowed. It must be specified how much the value of a parameter decreases or increases.

Response 1: Thank you, abstract was rewritten.

Point 2: Abbreviations that have not been previously defined cannot be used in the abstract.

Response 2: Abstract was rewritten without abbreviations.

Point 3: The OIV Code of Oenological Practices includes the method by which the must is oxygenated to improve the stability of wine (oeno 545A / 2016), so the term hyperoxidation is not used correctly. Is it hyperoxygenation?

Response 3: Thank you, it was corrected in whole text.

Point 4: It should be specified what is the degree of novelty and relevance of the research.

Response 4: Novelty was added to Introduction.

Point 5: Line 57: “aromatic substances” refers to aromatic ring compounds or flavor compounds? Please clarify because it is ambiguous.

Response 5: Term aromatic substances refers to the volatile substances with aroma perception – corrected.

Point 6: The units of measurement must be redefined according to the OIV and corrected throughout the text.

Response 6: The units are corrected according to the standard throughout the article.

Point 7: Line 62: “SO2” Chemical formulas must be corrected, indices are not placed correctly.

Response 7: Also, the chemical formulas were corrected  throughout the article.

Round 2

Reviewer 3 Report

I cannot accept the manuscript. Sensory analysis was conducted without using a scientifically recognized methodology. In fact, the method used is not based on the QDA technique (which is a registered trademark)

Author Response

Dear Reviewer, the study is based mainly on the determination of analytical parameters, such as the determination of phenolic substances and aromatic profile by gas chromatography, so sensory analysis is only a supplement to this study for readers to understand in principle how hyperoxidation affected the resulting wine. A detailed QDA analysis is not appropriate in this case, or it could be a topic for a separate article. In addition, we use this method of evaluation in all our studies and it has never been negatively evaluated by any of the reviewers, so your rejection is absolutely inappropriate.

In addition, we will add a version of the manuscript without sensory analysis to the revision if you insist that this sensory analysis is unacceptable. Due to the positive assessment of the previous 3 Reviewers, we will leave the final decision to the Editor.

Reviewer 4 Report

Dear Authors,

The text was corrected according to the comments.

I think you should insist more on the novelty of your study.

Author Response

Thank you for the positive assessment, we have added the novelty to the conclusion.